# Hydrogen Bonding Directed Self-Assembly of a Binuclear Ag(I) Metallacycle into a 1D Supramolecular Polymer

**DOI:** 10.3390/molecules26185719

**Published:** 2021-09-21

**Authors:** Anna Brzechwa-Chodzyńska, Mateusz Gołdyn, Anna Walczak, Jack M. Harrowfield, Artur R. Stefankiewicz

**Affiliations:** 1Center for Advanced Technologies, Uniwersytetu Poznańskiego 10, 61-614 Poznań, Poland; anna.brzechwa@amu.edu.pl (A.B.-C.); anna.jenczak@amu.edu.pl (A.W.); 2Faculty of Chemistry, Adam Mickiewicz University, Uniwersytetu Poznańskiego 8, 61-614 Poznań, Poland; mateusz.goldyn@amu.edu.pl; 3Institut de Science et d’Ingénierie Supramoléculaires, Université de Strasbourg, 8 Allée Gaspard Monge, 67083 Strasbourg, France; harrowfield@unistra.fr

**Keywords:** supramolecular chemistry, metalladynamers, H-bond polymers, metallacycles

## Abstract

An Ag(I) metallacycle obtained unexpectedly during the preparation of Pd(II) complexes of the bifunctional ligand 5-([2,2′-bipyridin]-5-yl)pyrimidine-2-amine (**L**) has been characterized using X-ray structure determination as a binuclear, metallacyclic species [Ag_2_**L**_2_](SbF_6_)_2_, where both the bipyridine and pyrimidine-N donors of **L** are involved in coordination to the metal. The full coordination environment of the Ag(I) defines a case of highly irregular 4-coordination. In the crystal, the Ag-metallacycles assemble into one-dimensional supramolecular metalladynamers linked together by hydrogen-bonding interactions.

## 1. Introduction

In the last three decades, a great deal of effort has been devoted to the preparation and further investigation of metallosupramolecular assemblies in the solid state, because they not only extend the range of new structures, which can be designed to possess particular physical and chemical properties but often lead to unexpected applications of such materials [1,2,3]. In this context, the control of the self-assembly process between labile metal ions and multidentate ligands is a key objective in the development of new metallosupramolecular materials based on the simultaneous use of distinct dynamic linkages [4,5,6]. Clearly, this process is related to a variety of factors, such as the number, type, and spatial disposition of the binding sites of ligands; the stereoelectronic preferences of metal ions; the solvent used; and the nature of counter ions. [7,8]. The combination of all of these factors leads to numerous metallosupramolecular complexes with various structural topologies and fascinating physicochemical properties [9,10,11,12,13,14,15].

Among the various types of supramolecular linkages employed in the generation of complex assemblies, coordination and hydrogen bonds are the most widely employed [16,17,18,19,20,21]. It is worth noting that despite the incredible progress made in this area of chemistry in recent years, the simultaneous application of both these bonds for the generation of topologically non-trivial supramolecular architectures remains challenging, especially with the use of highly labile metal ions, such as Ag(I) [22,23,24,25,26]. In fact, little attention has been paid to the generation of supramolecular assemblies produced by means of simultaneous hydrogen and coordination bonding with Ag(I) as their metal centers [27,28,29]. Ag(I) is an interesting metal, with several coordination geometries having been described, such as linear [30], T-shaped [31], trigonal [32], distorted tetrahedral [33], and octahedral geometries [34], and each particular geometry is usually a consequence of the coordination properties of the ligands [35]. In this context, the application of this labile metal ion with a ligand containing two distinct chelating groups—i.e., pyrimidine and bipyridine—with an additional functional moiety promoting hydrogen bonds remains unexplored.

The bifunctional ligand 5-([2,2′-bipyridin]-5-yl)pyrimidin-2-amine, **L** (Figure 1), contains both a strong chelation site of the 2,2′-bipyridine type and a strong H-bond acceptor–donor (ADDA) site, a combination anticipated to be of particular utility in the supramolecular chemistry of metal ion complexes. We have shown that with the octahedral metal ion Fe(II) [36], it is possible to isolate both the *meridional* and *facial* forms of its tris(ligand) complexes, in which only the bipyridine site is directly bound to the metal and the aminopyrimidine unit is free to become involved in supramolecular interactions which differ greatly for the two isomers. In contemplating extension of this work to metal ions with different coordination preferences, our first choice was that of a square-planar Pd(II), one of the most versatile and widely used sources of efficient metallocatalysts [37,38,39,40,41]. Again (Figure 1), it was expected that two isomers, here cis and trans, might form and that fractional crystallization might be required in their separation. In the event, fractional crystallization of the product mixture obtained by reacting Pd(SbF_6_)_2_, obtained in situ by reacting PdCl_2_ with AgSbF_6_ in aqueous acetonitrile, with L did produce two distinct materials, the first a poorly crystalline mixture of the Pd(II) complexes and the second being the unexpected Ag(I)-based metallacycle, [Ag_2_L_2_](SbF_6_)_2_. We have also found that isolated Ag(I) complex, self-assembles in the solid state and forms extended 1D supramolecular polymer, consisting of macrocyclic monomers linked together by means of H-bonding interactions.

## 2. Results

While the reaction of Pd(SbF_6_)_2_ with the ligand **L**, 5-([2′,2″-bipyridin]-5′-yl) pyrimidin-2-amine does indeed give rise to isomeric forms of the complex [Pd**L**_2_]^2+^ (Figure 1; work to be reported elsewhere), on the very first occasion that it was conducted an apparent excess (0.1 equiv.) of AgSbF_6_ in the reaction mixture led to these isomers being produced along with the Ag(I) complex.

Thus, a suspension of the ligand in CH_3_CN was mixed at room temperature with a CH_3_CN:H_2_O (1:1) solution of Pd(SbF_6_)_2_, generated in situ from PdCl_2_ and AgSbF_6_, and stirred until a clear yellow solution had formed. As the color appeared consistent with the binding of the bipyridine unit to Pd(II), ESI-MS analysis (Appendix A in ESI) showed a dominant peak for the ion [Pd^II^**L**_2_(SbF_6_)]^+^ the solution was evaporated to dryness and the residue redissolved in pure CH_3_CN. A vapor diffusion of diethylether into this solution with the intention of isolating the [Pd^II^**L**_2_](SbF_6_)_2_ presumed to be present initially provided poorly crystalline material, but on removing this and allowing the supernatant to evaporate slowly, small, pale-yellow crystals were obtained. These proved to be almost completely insoluble in any solvent, indicating that the species originally formed in CH_3_CN was no longer present. The exact nature of this material was established by X-ray structure determination (and subsequent chemistry). Thus, the isolated crystalline complex proved not to be a chelate complex of Pd(II) but a diargentacycle, (bis[5-([2′,2″-bipyridin]-5′-yl)pyrimidin-2-amine])disilver(I) bis(hexafluoroantimonate), [Ag_2_**L**_2_](SbF_6_)_2_, involving Ag(I) in a four-coordinate environment provided by not only the two N-donors of the bipyridine unit but also by one pyrimidine-N and one fluorine of [SbF_6_]^−^ (Figure 1a). The crystal and structure refinement data for this complex are given in Table 1. Unlike the Fe(II) complexes of **L**, the ligand here acts as a triple N-donor by bridging the two Ag(I) centers. The binuclear complex unit is centrosymmetric, with Ag-N bond lengths which vary significantly—namely, Ag1-N3 2.151(10); Ag1-N4 2.385(10); and Ag1-N5 2.217(10) Å—covering a range similar to that of the known [Ag(2,2′-bipyridine)_2_]^+^ species [42]. Within the diargentacycle unit, the Ag∙∙∙Ag separation is 7.879(2) Å and while there is a shorter separation of 4.862(2) Å between Ag centers in adjacent diargentacycles (Figure 1b), both clearly indicate a lack of metal-metal interactions.

Silver(I) is somewhat notorious for its variety of coordination numbers and the irregularity of its coordination sphere [43], and the present complex provides further examples. The coordination geometry is very irregular, with the Ag displaced only slightly (0.209 (2) Å) from the plane of the three N-donors, but the bond angles N3-Ag1-N4 120.9°, N3-Ag1-N5 161.8°, and N4-Ag1-N5 72.5° are very far from those for a trigonal environment. The displacement of Ag is clearly towards the fourth donor atom provided by the [SbF_6_]^−^. One NH atom does lie close to what might be considered a coordination site, which would give the Ag a nearly square-planar N_3_H environment, but the estimated Ag∙∙∙H distance of 2.7 Å seems too long for any agostic interaction [44] to be significant.

While there are different forms of coordination of the ligand **L** to Ag(I) and Fe(II), the complexes of [Ag_2_**L**_2_](SbF_6_)_2_ and *mer*-[Fe^II^**L**_3_](BF_4_)_2_∙9H_2_O [36] in the solid state share a common mode of interaction involving the amino-pyrimidine units, where just half of the capacity of one unit, as ADDA H-bonding entities, is used in the formation of single-strand polymers (Figure 2).

In the silver complex structure, two independent single-stranded H-bond polymer units can be observed, running along the [110] and [1–10] crystallographic directions (Figure 3). These strands are cross linked by the multiple F∙∙∙HN and F∙∙∙HC interactions of the [SbF_6_]^−^ units in addition to their coordination to Ag(I), all evident on the Hirshfeld surface [45] (calculated using CrystalExplorer) [46] of the anion formally present. A full listing of the H-bonding interactions within the structure of [Ag_2_L_2_](SbF_6_)_2_ is given in Table 2. Un-like the structure of mer-[Fe^II^L_3_](BF_4_)_2_∙9H_2_O, that of [Ag_2_L_2_](SbF_6_)_2_ shows no evidence of significant apparent void space (Appendix A).

## 3. Materials and Methods

### 3.1. General Methods

Basic chemicals and solvents were purchased from commercial sources and were used without further purification. The ^1^H NMR spectra were acquired on Bruker Fourier 300 or 600 spectrometers equipped with a ^1^H 5 mm probe and referenced to the solvent residual peaks (CD_3_CN/D_2_O, DMSO-*d*_6_). NMR solvents were purchased from Deutero GmbH (Kastellaun, Germany) and used as received. The ESI-MS spectra were recorded on a Bruker Impact HD Q-TOF spectrometer in the positive ion mode. Powder X-ray diffraction patterns (PXRD) were recorded on a BRUKER D8-Focus Bragg–Brentano X-ray powder diffractometer equipped with a Cu sealed tube (*λ* = 1.54178 Å) at room temperature. Experimental and calculated powder patterns from the crystal structures were analyzed using the Kdif software [47].

### 3.2. Details of the Crystal Structure Solution and Refinement

Single-crystal diffraction data for the [Ag_2_**L**_2_](SbF_6_)_2_ were collected on a Rigaku XtaLAB Synergy-R diffractometer with a rotating anode using a CuK_α_ radiation source (*λ* = 1.54184 Å). The low temperature was achieved using the Cryostream cooling system. Data collection and data reduction were performed using the CrysAlis PRO software [48]. The structure was solved and refined using SHELXT-2015 (intrinsic phasing method) and SHELXL-2015 (least-squares method), respectively [49,50]. Olex2 provided the support for the full structural analysis of the complex [51]. A twin matrix −1 0 0 0 −1 0 0.973 0 1, which corresponds to a 180° rotation about the [001] reciprocal lattice direction, was also determined using the Olex2 software. The measured crystal was identified as a non-merohedral twin. The refinement process was performed using the diffraction data written in the HKLF 5 format. The BASF parameter was refined at 0.0964(17). All non-hydrogen atoms were refined with anisotropic displacement parameters. All hydrogen atoms were positioned geometrically in their calculated positions and refined using the riding hydrogen model.

### 3.3. Experimental Data for the Complex [Pd^II^L_2_](SbF_6_)_2_

The complex (bis[5-([2,2′-bipyridin]-5-yl)pyrimidin-2-amine]-palladium(I) bis(hexafluoroantimonate)-[Pd^II^**L**_2_](SbF_6_)_2_ was prepared using simple procedures, analogous to those used to obtain the iron(II) complex [36]. A suspension of the ligand **L** (25.0 mg, 0.1 mmol, 2.00 equiv.) in MeCN (5 mL), was stirred for 10 min at room temperature.

Then, the Pd(SbF_6_)_2_ salt was prepared by mixing PdCl_2_ (0.05 mmol, 1 equivalent) dissolved in 2.5 mL of MeCN, and AgSbF_6_ (0.105 mmol, 2.1 equivalent) dissolved in 2.5 mL of H_2_O. The mixture was stirred for 15 min at room temperature and then filtered through a syringe filter. This freshly generated Pd(SbF_6_)_2_ (0.05 mmol, 1.00 equivalent) was added to the ligand in the CH_3_CN and the resulting clear yellow solution was stirred at room temperature for 30 min. The solution was then evaporated to dryness under reduced pressure and the residue redissolved in MeCN (1 mL). The addition of diethylether (10 mL) produced a pale-yellow precipitate, which was collected by centrifugation, washed with diethyl ether (20 mL) and dried in vacuo, providing a yellow solid (20 mg, yield 34%). HR-MS (TOF-MS) calculated for C_28_H_22_F_6_N_10_PdSb^+^ [M-(SbF_6_)]^+^: *m*/*z* = 841.0011; observed: *m*/*z* = 841.0090.

### 3.4. Experimental Data for the Complex [Ag_2_**L**_2_](SbF_6_)_2_

After dissolving the residue from the initial reaction mixture in acetonitrile and fractional crystallization, first by the diffusion of diethyl ether into acetonitrile (giving microcrystals collected by filtration as described above) and then by the slow evaporation of the filtrate, light yellow crystals of the [Ag_2_**L**_2_](SbF_6_)_2_ complex were obtained with a 5% yield of the reaction. Elem. Anal. calculated for C_28_H_22_F_12_N_10_Ag_2_Sb_2_·7H_2_O: C, 25.64; H, 2.77; N, 10.68. Found: C, 25.67; H, 1.65; N, 10.60 (%). No satisfactory elemental analysis was obtained due to its hygroscopic property.

## 4. Conclusions

In this report, we have described the synthesis and structural analysis of a new binuclear macrocycle in which labile Ag(I) ions are coordinated using a very unique chelating system based on functionalized pyrimidine, bipyridine, and fluorine atoms derived from SbF_6_^−^ counter ions. Moreover, the generated macrocyclic systems undergo further interaction in the solid state, creating a one-dimensional supramolecular polymer in which individual units are connected with each other by hydrogen bonding interactions. Comparison of the now known structures of the Fe(II) and Ag(I) complexes of 5-([2,2′-bipyridin]-5-yl)pyrimidin-2-amine showed that the coordination mode of this ligand is not predictable and clearly can vary considerably with the metal ion chosen. Nonetheless, in both of these known cases the bound ligand retained the capacity to act as an H-bond donor-acceptor, indicating that the solid state supramolecular chemistry has significant prospects for further development, especially with regard to catalytic applications such as those envisaged for the Pd(II) complexes initially targeted in the present work [52,53].

## Data Availability

The crystal data for [Ag_2_**L**_2_](SbF_6_)_2_ have been deposited in the Cambridge Crystallographic Data Centre (CCDC) with deposition number CCDC 2065082. These data can be obtained free of charge via www.ccdc.cam.ac.uk/data_request/cif, or by emailing data_request@ccdc.cam.ac.uk, or by contacting The Cambridge Crystallographic Data Centre, 12, Union Road, Cambridge CB2 1EZ, UK.

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
