# Peer review of "Hydrogen Bonding Directed Self-Assembly of a Binuclear Ag(I) Metallacycle into a 1D Supramolecular Polymer"

_molecules, 2021, doi:10.3390/molecules26185719_

Round 1

Reviewer 1 Report

Recommendation: Publish after revision required

The manuscript Molecules-1381165 describes the synthesis of a metallacycle of silver with the general formula [M2L2][X]2 ( M = Ag(I), L = 5 - ([2',2"-bipyridin]-5'-yl) pyrim-idin-2-amine) and X = SbF6). The metallacycle was obtained accidentally but reproducibly upon treatment of AgSbF6 and PdCl2 in the presence of L. The molecular structure of [Ag2L2][SbF6]2 was determined and showed the formation of the metallacycle. Every Ag(I) is coordinated by two L ligands in a distorted trigonal planar fashion, in addition the solid-state structure shows a weak interaction with the SbF6 counter anion providing a four-coordinate environment. The crystal packing reveals that the individual metallacycles interact further through hydrogen bonding via the free amine function of L to generate a coordination polymer.

Overall I feel the science is sound and the paper is well presented. However there are several papers related to this chemistry that were not mentioned and should be included in the revised version before I recommend its publication in Molecules.

1. The following references related to the topic should be in the revised version: 

1) Supramolecular Chemistry, 2018, 30, 822. 2) Chem. Soc. Rev. 2013, 42, 1619-1636 3) Chem. Rev. 2012, 112, 2015. 4) Eur. J. Inorg. Chem. 2011, 4558.

2. The counter ion seems to interact with the metallacycle at least in the solid state. Have the authors tried to study the effect of other anions (BF4, CF3SO3, NO3) on the self-assembly?

Author Response

Reviewer 1:

Comments R1_1: The manuscript Molecules-1381165 describes the synthesis of a metallacycle of silver with the general formula [M2L2][X]( M = Ag(I), L = 5 - ([2',2"-bipyridin]-5'-yl) pyrim-idin-2-amine) and X = SbF6). The metallacycle was obtained accidentally but reproducibly upon treatment of AgSbF6 and PdCl2 in the presence of L. The molecular structure of [Ag2L2][SbF6]2 was determined and showed the formation of the metallacycle. Every Ag(I) is coordinated by two L ligands in a distorted trigonal planar fashion, in addition the solid-state structure shows a weak interaction with the SbF6 counter anion providing a four-coordinate environment. The crystal packing reveals that the individual metallacycles interact further through hydrogen bonding via the free amine function of L to generate a coordination polymer.

Overall I feel the science is sound and the paper is well presented. However there are several papers related to this chemistry that were not mentioned and should be included in the revised version before I recommend its publication in Molecules.

Answer R1_1We would like to thank the Reviewer for appreciating our work.

Comments R1_1: The following references related to the topic should be in the revised version: 1) Supramolecular Chemistry, 2018, 30, 822. 2) Chem. Soc. Rev. 2013, 42, 1619-1636 3) Chem. Rev. 2012, 112, 2015. 4) Eur. J. Inorg. Chem. 2011, 4558.

Answer R2_1We would like to thank the Reviewer for this comment and indication of literature that perfectly complements the idea of our research.

Our Action on R2_1The following reference: Supramolecular Chemistry, 2018, 30, 822 (placed as Ref. 28 in the revised version of the manuscript); Chem. Soc. Rev. 2013, 42, 1619-1636 (placed as Ref. 6 in the revised version of the manuscript); Chem. Rev. 2012, 112, 2015 1636 (placed as Ref. 3 in the revised version of the manuscript); Eur. J. Inorg. Chem. 2011, 4558 (placed as Ref. 29 in the revised version of the manuscript) have been added in the revised version of the manuscript.

Comments R3_1: The counter ion seems to interact with the metallacycle at least in the solid state. Have the authors tried to study the effect of other anions (BF4, CF3SO3, NO3) on the self-assembly?

Answer R3_1We agree with the Reviewer that anions may play important role in this assembly. For the purpose of this particular project we did not try complexation reactions with other counterions as our main goal was to introduce maximum complex solubility using SbF6- counterion. However, our results have shown that the use of even these counterions did not significantly improve the solubility of the obtained complex, therefore we suspect that the use of other counterions, e.g. ClO4-, could cause even greater problems with solubility. Nevertheless, we think it may be a good introduction to further research.

Reviewer 2 Report

The paper by Stefankiewicz and coworkers reports on the unexpected formation of binuclear metallocyclic species [Ag2L2](SbF6)2 during the preparation of  Pd(II) complexes of the bifunctional ligand 5-([2,2'-bipyridin]-5-yl)pyrimidine-2-amine (L).

The Ag-metallocycles undergo further interaction in the solid state affording a one-dimensional supramolecular polymer whose units were linked together by hydrogen-bonding interactions. The structure of the newly synthesized compounds was investigated through X-ray analysis.

The work is well-written and correctly presented, and can be accepted for publication in Molecules, but some improvements are required regarding the preparation and characterization of the final compounds.

  • In the main text (lines 71-73) authors report that the Ag(I) complex was obtained during the attempts to prepare complex [PdL2]2+ by using an “apparent excess” of AgSbF6. I think this data is too generic and deserves further investigation. Did the authors try to repeat the experiment using first an equimolar amount and then a gradually increasing amount of AgSbF6? This is particularly important to maximize the yield in [Ag2L2](SbF6)2 (why the yield is not reported in the experimental section?). I believe that a reliable preparation of the complex [Ag2L2](SbF6)2 must definitely be reported in the experimental part.
  • It is not clear to me why the elemental analysis of [Ag2L2](SbF6)2 has been calculated in the presence of water (7H2O). Please explain.
  • The observed mass of [PdIIL2(SbF6)] leaves me perplexed. The difference between calculated and observed mass is – 0.008 m/z unit (9 ppm). The mass error is far too large for a high-resolution data. An observed value within 0.003 m/zunit of the calculated value of a parent‑derived ion is usually requested for supporting a molecular formula for compounds with molecular masses below 1000 amu. Authors should check the purity of the compound and repeat the HRMS experiment.
  • The Instruction for Authors state that “complete characterization data must be given for all new compounds”. Therefore, both, proton and carbon NMR data should be included for [PdIIL2(SbF6)] and [Ag2L2](SbF6)2.

Minor remarks:

  • Lines 32–34: “The combination of all of this (these) factors leads to numerous metallosupramolecular complexes with various structural topologies and fascinating physicochemical properties”. To justify this statement, the authors report 4 bibliographic references, three of which are their papers. Since the examples are "numerous" I suggest not to focus only on their works, but to report other bibliographic references.
  • The same thing can be said for the following statement: lines 35–37 “Among the various types of supramolecular linkages employed in the generation of complex assemblies, coordination and hydrogen bonds are the most widely employed”, where two of the three bibliographic references indicated are authors’ papers.
  • Line 129: why two N4-Ag1-N5 bond angles are reported? What is the correct value, 72.5° or 72.9°?
  • Lines 214–218: “Nonetheless, in both of these known cases the bound ligand retains a capacity to act as an H-bond donor-acceptor, indicating that the solid state supramolecular chemistry has significant prospects for further development, especially in regard to catalytic applications such as are envisaged for the Pd(II) complexes initially targeted in the present work”. On what basis do the authors make this consideration? They should explain better using (and reporting es references) examples from literature

Reviewer 3 Report

Artur R. Stefankiewicz and co-workers have very nicely described the unexpected crystallization of Ag(I) metallocycle during the preparation of Pd(II) complexes. This article can be acceptable for publications, however, following minor comments should to be addressed.

  1. Authors should incorporate the in-detailed description of synthesis of in situ generated from Pd(SbF6)2 from PdCl2 and AgSbF6.. It’s really important because Ag+ ions are still present in the residue which leads to the crystallization as [Ag2L2](SbF6)2 instead of expected [PdIIL2](SbF6)2.
  2. In section 3.3, authors explained if the residue dissolved in ACN was treated with Et2O furnished a pale yellow precipitate which showed the formation of [PdIIL2](SbF6)2. Did they observe any existence of [Ag2L2](SbF6)2 as a minor component? Since the NMR spectrum of this in Figure S1 is broad, did they face any solubility issue (for instance partial solubility) while dissolving the separated solid product of [PdIIL2](SbF6)2 in the solvent? In NMR overlap, the NMR spectra of free L also need to be included to investigate the changes in chemical shift.
  3. To investigate the role of Pd, they might try to prepare a complex by reacting ligand (L) directly with AgSbF6 to see whether they would get the insoluble complex [Ag2L2](SbF6)2.
